# Repurposing of Commercially Existing Molecular Target Therapies to Boost the Clinical Efficacy of Immune Checkpoint Blockade

**DOI:** 10.3390/cancers14246150

**Published:** 2022-12-13

**Authors:** Debottam Sinha, Philip Moseley, Xuehan Lu, Quentin Wright, Brian Gabrielli, Ian H. Frazer, Jazmina L. G. Cruz

**Affiliations:** 1Faculty of Medicine, Frazer Institute, The University of Queensland, Brisbane, QLD 4102, Australia; 2Mater Research Institute, The University of Queensland, Brisbane, QLD 4102, Australia

**Keywords:** novel treatment, hostile tumor microenvironment, immunosuppression, cytotoxic T cells, small molecule inhibitors

## Abstract

**Simple Summary:**

Epithelial cancers, such as lung, breast, and colon cancers, have high mortality rates because of their ability to spread across multiple organs in the body. Besides the standard of care which includes chemotherapy and radiotherapy, approaches directed to use patient’s own immune responses against the disease called immunotherapies have emerged as a powerful treatment option. In the past 10 years, immune checkpoint blockade, a form of immunotherapy which either stimulates or removes the breaks of the immune response against cancer, is having the largest impact in the clinic. However epithelial cancers are commonly either naturally resistant or develop resistance to these types of treatments. Hence, there is an urgent need to boost the effectiveness of immune checkpoint blockers. Small molecule inhibitors are chemical molecules which are specifically designed to target important cancer proteins and unlike chemotherapy, typically have manageable toxicity. These inhibitors have shown good efficacy in reducing tumour growth but more recently, they have been shown to enhance the performance of immune cells in eliminating cancers. In this review, we have focused on tactical usage of small molecule inhibitors to boost the efficacy of immune checkpoint blockers. We believe our review will pave the way for novel research combining the two therapeutic modalities.

**Abstract:**

Immune checkpoint blockade (ICB) is now standard of care for several metastatic epithelial cancers and prolongs life expectancy for a significant fraction of patients. A hostile tumor microenvironment (TME) induced by intrinsic oncogenic signaling induces an immunosuppressive niche that protects the tumor cells, limiting the durability and efficacy of ICB therapies. Addition of receptor tyrosine kinase inhibitors (RTKi) as potential modulators of an unfavorable local immune environment has resulted in moderate life expectancy improvement. Though the combination strategy of ICB and RTKi has shown significantly better results compared to individual treatment, the benefits and adverse events are additive whereas synergy of benefit would be preferable. There is therefore a need to investigate the potential of inhibitors other than RTKs to reduce malignant cell survival while enhancing anti-tumor immunity. In the last five years, preclinical studies have focused on using small molecule inhibitors targeting cell cycle and DNA damage regulators such as CDK4/6, CHK1 and poly ADP ribosyl polymerase (PARP) to selectively kill tumor cells and enhance cytotoxic immune responses. This review provides a comprehensive overview of the available drugs that attenuate immunosuppression and overcome hostile TME that could be used to boost FDA-approved ICB efficacy in the near future.

## 1. Introduction

One of the hallmarks of cancer is the ability of malignant cells to avoid immune surveillance. Tumors subvert many different normal immunosuppressive mechanisms to either block detection or suppress immune recognition. One mechanism that has proven to be targetable and enhances immune recognition is the immune checkpoint pathways. Research on regulation of normal immune responses has identified inhibitory receptors on immunocytes whose normal function is to limit healthy immune responses or block auto-immune detection and response. Antibodies that block inhibitory receptors can enhance and prolong immune responses. This research has enabled development of immune checkpoint blockade (ICB) monoclonal antibodies (mAbs) that improve clinical outcome for multiple cancers, notably metastatic melanoma, for which the 5 years survival rate can reach up to 44% with nivolumab treatment, and 26% for ipilimumab treatment [1,2]. Currently, Food and Drug Administration (FDA) approved ICB monoclonal antibodies (mAbs) in use in the clinic are: (i) ipilimumab (Yervoy^®^) blocking cytotoxic T-lymphocyte–associated antigen 4 (CTLA-4); (ii) cemiplimab (Libtayo^®^), (iii) nivolumab (Opdivo^®^) and (iv) pembrolizumab (Keytruda^®^) inhibiting programmed death-1 (PD-1); (v) atezolizumab (Tecentriq^®^), (vi) avelumab (Bavencio^®^), and (vii) durvalumab (Imfinzi^®^) targeting programmed death ligand-1 (PD-L1); and very recently approved (March, 2022) (viii) relatlimab (Opdualag^®^), targeting lymphocyte activation gene-3 (LAG-3) and (ix) tiragolumab (Tecentriq^®^) against T cell immunoglobulin and immunoreceptor tyrosine-based inhibitory motif (ITIM) domain (TIGIT) [3,4,5,6,7]. Additionally, targeting other immune checkpoint molecules such as T cell immunoglobulin and mucin-domain containing-3 (TIM-3), Signal regulatory protein α (SIRP α) and V-domain immunoglobulin (Ig) suppressor of T cell activation (VISTA) using mAbs are also currently under evaluation at either at preclinical or early phase of clinical trials [8,9,10].

The immune checkpoints (namely PD-1, CTLA-4, LAG-3, TIM-3, TIGIT) curtail overstimulation of the immune system post antigen exposure to restore normal homeostasis and avoid exacerbated immune responses. This balance is maintained by binding of these inhibitory receptors expressed by several immune subsets including T cells or NK cells, with complementary co-stimulatory ligands expressed by antigen presenting cells (APCs) and other myeloid cells, respectively. Interestingly, in cancer, tumor cells upregulate the expression of PD-L1 which binds PD-1 with high affinity resulting in the inactivation of Zeta-chain-associated protein kinase 70 (ZAP70) and CD28 and subsequent TCR signalling cascade inhibition. CTLA-4 competes with T-cell activation receptor CD28 for binding to CD80 and CD86 (co-stimulatory molecules). These receptors are highly expressed by cancer cells and upon interaction with CTLA-4 results in reduction in T cell proliferation and interleukine-2 (IL-2) production [11]. Other checkpoint molecules bind to their respective targets expressed on cancer cells to trigger immune malfunction and to facilitate immune evasion. Therefore, the rationale behind inhibition of these checkpoint interactions through engineered ICB mAbs is to override immunosuppression facilitating reactivation of the adaptive immune response [3]. However, apart from melanoma, the response rates to ICB mAbs across a variety of tumor types have generally been less than 30% and face a stiff challenge in clinic [12] (Table 1).

Immunotherapy for epithelial cancers can fail because there is an immunosuppressive tumor microenvironment (TME) as reviewed by de Miguel M, et al. [30]. Development of resistance to therapy, with local or metastatic tumor recurrence, occurs with ICB mAbs, as it does for small molecule drugs targeting cell tumor metabolism [31]. Hence, understanding how cancer cells influence the local immune environment and how small molecule cytotoxic cancer therapies can improve tumor immunogenicity is essential to design new strategies that can enhance the therapeutic effect of ICB mAbs. In this review, we focus on several anti-cancer drugs that also modulate the immune anti-tumor response and could be strategically repurposed in combination with FDA-approved ICB mAbs to improve clinical outcomes.

## 2. Mechanism Driving Resistance to ICB

The TME consists of a heterogeneous population of cells that collectively contribute pro-immune and immunosuppressive signalling as shown in Figure 1. Macrophages, myeloid-derived suppressor cells (MDSCs), regulatory T cells (T_regs_), and cell-free factors including anti-inflammatory cytokines are major contributors to local immunosuppression. These factors generate a protective shield to defends the tumor from cytotoxic T cells by inhibiting their T cell trafficking and proliferation. This can be achieved by suppressing neoantigen or tumor associated antigen presentation, or by inducing T cell exhaustion. As these processes have been extensively reviewed, we will briefly highlight their importance in driving resistance to ICB [32,33]. T cell trafficking comprises a step wise process of rolling, adhesion, extravasation, and chemotaxis, governed by pro- and anti-inflammatory chemokines (chemokine (C-C motif) ligand (CCL) 5, CCL17, CCL22, chemokine (C-X-C motif) ligand (CXCL) 8, and CXCL12) and cytokines including interleukin (IL)-4, IL-6, IL-10, IL-11, and IL-13. This cocktail of secreted molecules favours mobilization of MDSCs and T_regs_ hindering the recruitment of cytotoxic T cells to the tumor [34]. Similarly, impaired interferon (IFN)-γ signalling accompanied by suppression of dendritic cell (DC) maturation and recruitment leads to hindered T cell proliferation and priming [35].

Additionally, impaired adaptive immune responses can result from reduced tumor antigen presentation, a consequence of downregulation of expression of major histocompatibility complex (MHC)-I on cancer cells [36]. Post-translationally, loss of β2-microglobulin is the major contributor to disruption of MHC-I folding and transport to the cell surface [37]. Interestingly, mutations within the T cell receptor binding domain of MHC, reported in colorectal cancer, block immunosurveillance by abrogating cytotoxicity [38]. Intrinsically, immunologically cold tumors, such as pancreatic and triple negative breast cancers, have low tumor mutational burden (TMB), limiting presentation of immunogenic neoantigens, and thus tumor specific cytotoxic T cell repertoires [39,40]. Over ten years of ICB’s use in the clinic, have shown that tumors lacking neoantigen presentation have poor treatment outcome, whereas tumors with high TMB and neoantigen presentation, including melanoma, have an improved ICB response [41].

Additional immunosuppressive mechanisms employed by cancer cells include expression of immunomodulatory ligands (PD-L1, CD47 and CD155) that bind to corresponding cytotoxic T cells receptors (PD-1, SIRPα and TIGIT). Persistent signalling induced by these interactions leads to T cell exhaustion. Though ICB mAbs should bypass these mechanisms, compensatory upregulation of other checkpoint pathways, such as lymphocyte-activation gene 3 (LAG-3) or CTLA-4 can follow [42]. Activation of multiple checkpoint signalling networks can contribute to treatment failure after prolonged ICB cancer treatment, highlighting the need for combination strategies to minimize drug resistance and maximise the durability and efficacy of ICB mAbs.

## 3. Repurposing SMIs to Improve Efficacy of ICB

Targeted therapies using small molecules (SMIs) known to inhibit molecular or biochemical pathways critical for tumor growth and maintenance also have an impact on tumor infiltrating immune effector cells. Combinations of SMIs with immune checkpoint blockade has proven to be effective in pre-clinical models, and several clinical trials are underway (Table 2) [43]. Therapies targeting receptor tyrosine kinases (RTK) can induce immunogenic modulation either by improving the cytotoxic function of the adaptive immune system, or by blocking expression of immunosuppressive molecules such as PD-L1, to enhance T cell mediated elimination of cancer cells. In addition, RTK inhibitors increase frequency and function of effector immune cells in the TME of epithelial cancers (i.e., melanoma and colon cancer) while decreasing the number and function of immune suppressor cells [44,45].

SMIs can target RTKs such as Epidermal Growth Factor Receptor (EGFR), v-raf murine sarcoma viral oncogene homolog B1 (BRAF), KIT, Human Epidermal growth factor Receptor-2 (HER-2), phosphatidylinositol-4,5-bisphosphate 3-kinase (PIK3CA)/AKT/mammalian Target Of Rapamycin (mTOR) and Anaplastic Lymphoma Kinase (ALK). EGFR inhibitor-based therapies (Sunitinib, axitinib, erlotinib, gefitinib, imatinib) can influence T cell priming, increasing memory and effector T cell phenotypes. EGFR inhibitors can also impact on T cell tumor antigen recognition, activation, and trafficking of immune cells into the tumor. Additionally, they can sensitize cancer cells to immune effector cell mediated killing and antagonize cancer-induced immune suppression [57]. EGFR inhibitors also augment DCs function and tumor antigen presentation, enabling better T cell-mediated tumor destruction [57].

SMIs blocking immune check points can directly remodel the immune response in the TME. The blockage of the innate immune checkpoint CD47/SIRPα pathway using either antibodies or SMIs has been extensively investigated [58,59,60]. However, CD47 is a ubiquitously expressed cell surface protein also found on red blood cells and antibodies against CD47 have been associated with adverse events including anaemia [60]. Unlike ICBs, SMIs targeting CD47 displayed less toxicity as they either disrupt CD47/SIRPα interaction or modulate CD47 at the transcriptional, translational, and post-translational modification levels [60]. A good example is the SMI RRx-001 which skews the phenotype of tumor infiltrating macrophages from immune-suppressive M2 to highly phagocytic M1 [61]. In contrast to anti-CD47 antibodies, RRx-001 showed no hematologic toxicities in 9 clinical trials (~300 patients involved) and has positively progressed to Phase III (NCT03699956 and NCT02489903) [62,63]. As the majority of ICBs harness the power of adaptive immunity, the combination of these agents with innate immunity modulator such as RRx-001 is a very attractive approach that will need further evaluation with pre-clinical studies.

Although, effective LAG-3 SMIs are lacking promising development, recent research revealed that SMIs blocking glycogen synthase kinase-3 (GSK-3), such as SB415286 and elraglusib, not only down-regulated PD-1 expression enhancing CD8^+^ T cell cytotoxicity, but also reduced LAG-3 levels on T cells in mice [64]. Interestingly, the combination of GSK-3 SMI with anti-LAG-3 mAbs had a synergistic effect and was more effective than the SMI monotherapy alone in a melanoma mouse model [65,66].

TIGIT has proven to be of clinical interest due to its dual expression on tumor and immune cells, such as NKs and CD8^+^ tumor infiltrating T cells, and the positive correlation between TIGIT levels and PD-1 expression on human melanoma infiltrating CD8^+^ T cells [67,68]. In addition to anti-TIGIT antibodies, FDA-approved TIGIT SMIs liothyronine and azelnidipine are approved by the FDA with ability to block the interaction between TIGIT and CD155 [69,70]. Although liothyronine did not inhibit tumor cell proliferation in vitro, it significantly abrogated tumor growth in an in vivo model of colon adenocarcinoma (MC-38) by increasing the levels of CD8^+^ T cells within the tumor, protection that was lost when either CD4^+^ or CD8^+^ T or NK cells were depleted [70]. Interestingly, azelnidipine inhibits both CD47/SIRPα and TIGIT/PVR pathways by binding SIRPα and CD155, to enhance macrophage phagocytic activity and increase the infiltration of CD8^+^ T cells in murine MC-38 tumors. Although both TIGIT SMIs showed encouraging effects in tumor-bearing mice, their potential off target effects in endocrine system or ion channels should be carefully monitored in future clinical trials [69,70]. The recent FDA-approval of relatlimab (anti-LAG3, Opdualag^®^) in combination with nivolumab and tiragolumab (anti-TIGIT, Tecentriq^®^) to treat metastatic melanoma and NSCLC patients, respectively. This opens the door for new combination therapies with some of the previously mentioned FDA-approved ICB and SMIs that have the potential to synergistically reverse immune suppression and enhance anti-tumor response.

The mitogen-activated protein kinase (MAPK) signalling pathway is critical for tumor cell growth, proliferation, invasion, and metastasis in multiple cancers [71,72]. Mitogen Expressing Kinase (MEK) inhibitors (PD098059, trametinib and cobimetinib) were the first drugs developed to suppress the MAPK pathway. However, despite their high potency and selectivity, clinical response to MEK inhibitors as a single agent was largely disappointing [73]. Recent studies have shown the potential of MEK inhibitors for use in immune-sensitization by up-regulation of tumor antigen expression and presentation [74,75], and through production of IL-8 and vascular endothelial growth factor (VEGF), enhancing recruitment of immune cell to the tumor site [76]. Notably, Kang et al. [77] demonstrated in human NSCLC that trametinib (MEK1/2 inhibitor) enhances MHC-class I expression via signal transducer and activator of transcription-3 (STAT3) activation and upregulates chemokines associated with T cell infiltration and homing. Interestingly, a recent study, using a murine syngeneic BRAFV600E melanoma model, demonstrated improved efficacy of PMEL (premelanosome protein)-1-specific adaptive cell therapy, when combined with the BRAF + MEK inhibitors dabrafenib and trametinib [78]. The triple combination increased tumor T cell infiltration, leading to complete tumor regression [78]. Also recently, experiments using a head and neck squamous cell carcinoma (HNSC) model demonstrated that trametinib delays tumor initiation and progression by enhancing CD8^+^ T cell antitumor function and promoting development of long-term memory cells when combined with anti-PD-1 [79]. Clinical trials in which RTK inhibitors have been combined with ICB mAb therapy have shown promising response (Table 2). Other SMIs with immunomodulatory capacities, discussed below, are yet to be tested in combination with ICB mAb therapy and could also prove to be effective cancer therapeutics.

## 4. Taking Advantage of Cell Cycle Inhibitors

Deregulation of the cell cycle is a well-known hallmark of tumorigenesis and to date, multiple SMIs have been designed to target major players known to modulate this pathway in the cancer setting. The most interesting SMIs are designed to target the aberrant activity of CDK4/6 (FDA approved palbociclib, ribociclib, and abemaciclib). Overexpression of Cyclin D1 (the binding partner of CDK4/6) alongside loss of function of p16^INK4a^ (the endogenous CDK4/6 inhibitor), enables abnormal function of CDK4/6 leading to compromising the G1/S checkpoint of the cell cycle [80]. Though CDK4/6 inhibitors have been extensively utilised for the treatment of hormonal breast cancer, recent studies in melanoma (using mouse models) have highlighted their complementary immunotherapeutic activity [81,82,83,84,85]. Palbociclib has been shown to improve the anti-tumor efficacy of anti-PD-1/PD-L1 ICBs by enhancing MHC-I expression through type III interferon production. This drug reduces PD-L1 expression in mouse breast cancer cells and increases tumor cell production of T cell stimulants, such as CXCL10 and CXCL13 chemokines resulting in an increased lymphocyte recruitment within the TME [85,86,87]. In addition, CDK4/6i can act directly on T cells by diminished T_reg_ proliferation and enhancing effector T cell activity through downregulation of nuclear factor of activated T cells (NFAT), that regulates transcription in T_regs_ [81,82,88]. In mouse model, breast tumors treated with CDK4/6i showed an enhancement of stem or memory-like cytotoxic CD8^+^ T cells responsible for sustained clinical responses to ICB [86].

WEE1, an important regulator of G2/M phase of the cell cycle, has been recently shown to play an important role in dictating anti-tumor immune responses in preclinical small cell lung cancer models. Using AZD1775 (WEE1 inhibitor), Taniguchi et al. [89] demonstrated that WEE1 inhibition led to activation of the stimulator of interferon genes (STING)-TANK binding kinase (TBK)-interferon regulatory factor (IRF3) pathway which increased production of type I interferons (IFN-α and IFN-β) alongside pro-inflammatory chemokines (CXCL10 and CCL5). Furthermore, WEE1 inhibition triggered upregulation of STAT1 which induced upregulation of PD-L1 and IFN-γ expression, but upon combination with anti-PD-L1 blockade induced anti-tumor immune response in a CD8^+^ T cell dependent-manner [89].

Polo like kinase 1 (PLK1) is an important player in the regulation of the mitotic phase of the cell cycle and its expression is deregulated during tumorigenesis. PLK1 overexpressing tumors (most epithelial cancers) have been shown to have minimal tumor infiltrates alongside low MHC-I expression [90]. Metadata analysis of publicly available genomic data from TCGA (The Cancer Genome Atlas Program) dataset on 33 different cancer types patients who were treated with PLK1 inhibitor demonstrated increased anti-tumor immunity characterized by an upregulated expression of NK (natural killer)-cell-like gene signatures and genes involved in antigen presentation such as Transporter associated with antigen processing 1 & 2 (TAP1 and TAP2) [90]. In another study using preclinical NSCLC mouse model, PLK1i (BI2536) enhanced DC maturation and T cell infiltration [91]. Aurora Kinase A (AURKA) is an upstream regulator of PLK1 and regulates centrosome maturation and spindle formation in mitosis. Interestingly, in a recent study using a murine mammary tumor model, Alisertib (AURKA inhibitor) in combination with anti-PD-L1 therapy induced tumor regression. This combination was associated with reduced numbers of tumor-promoting myeloid cells (induced apoptosis of MDSCs) alongside significant increases of active CD8^+^ and CD4^+^ T cells [92].

KRAS (Kirsten rat sarcoma), is an important oncogene and its mutation is known to drive abnormal cell cycle progression and tumorigenesis in NSCLC, pancreatic ductal adenocarcinoma, and colorectal cancer (CRC). The common missense mutations observed in KRAS oncogene are: G12, G13, and Q61 and have been extensively investigated for designing targeted therapies [93]. Ostrem et al. [94] identified docking pocket in the KRAS-G12C mutant paving way for designing multiple covalent inhibitors. AMG 510 (sotorasib) was the first drug candidate which demonstrated success in clinical trials for KRAS-mutant cancers, especially NSCLC patients with KRAS-G12C mutation (32.2% achieved objective response and 88.1% achieved disease control) [95,96,97]. Consequently, it received fast track FDA approval for treatment of NSCLC patients harbouring KRAS-G12C mutations. Interestingly, these patients have a high response rate to ICBs compared with NSCLC patients with other mutations, such as EGFR [97]. Currently, sotorasib either alone or in combination with chemotherapy and ICB is under clinical trial (NCT04625647, NCT04185883) and could prove to be highly beneficial in inducing antitumor immunity in NSCLC patients.

## 5. Potential Application SMIs against DNA Damage Regulatory Proteins

Conventional chemotherapies and DNA damage response inhibitors (DDRi) both increase the load of DNA damage in tumor cells triggering an innate immune response, but also promote immunosuppressive signals [98,99]. However, conventional chemotherapies, through their less targeted approach, also kill immune cells and thus are poor candidates to combine with immunotherapies. Therefore, it is proposed that DDRis have fewer healthy tissue toxicities as they target tumor-specific defects and thus represent better candidates for combination with immunotherapies. The prototypic tumor targeted DDRis are the poly ADP ribosyl polymerase (PARP) inhibitors (PARPi), specially Olaparib which is currently under clinical trial in combination with ICB (Table 2). These drugs were identified as synthetic lethal interactors initially with BReast CAncer gene 2 (BRCA2) mutations, but since have been shown to have similar synthetic lethal interaction with any mutation that results in defective homologous recombination repair (HRR) [100]. However, clinical experience suggests that germline or somatic mutations of only a subset of HRR genes including *BRCA1/2* and Partner and localizer of BRCA2 (*PALB2*) confer sensitivity to PARPi, (olaparib) in patients [101]. One of the outcomes of olaparib treatment is increased DNA damage which triggers an innate immune response, commonly through the cGAS-STING pathway (cyclic-GMP-AMP synthase cGAS—Stimulator of Interferon genes) [102]. This can produce improved immune recognition that is further enhanced with ICB, although the effect appears to be independent of the functional status of HRR [103,104,105]. This innate immune response can be triggered by any agent that promotes DNA damage and can utilise either the canonical cGAS-STING or non-canonical pathways [106,107,108]. However, DNA damage can also trigger immunosuppressive responses such as upregulation of PD-L1 [109,110].

Ataxia telangiectasia and Rad3-related protein (ATR) and Checkpoint Kinase 1 (CHK1) are components of the cellular response to replication stress [111,112]. Although SMIs targeting ATR and CHK1 (M6620 (VX-970) and SRA737) have limited activity in patients as single agents [113,114], they have been shown to trigger innate immune signalling and can be combined with ICB to enhance anti-tumor immune responses in preclinical models and recently in clinical trials [115,116,117]. This may be a consequence of the ability of ATRi to block the DNA damage-induced expression of PD-L1 [109,110]. ATRis (AZD6738 and CHK1i (GDC-0575) also synergise with drugs that promote replication stress such as gemcitabine and cisplatin. However, when combined with standard doses of these drugs, they were associated with high levels of severe adverse haematological responses limiting their ability to be used with immunotherapies [111,118,119]. It is possible to avoid these adverse responses by using subclinical doses of replication stress promoting drugs such as gemcitabine or hydroxyurea in combination with CHK1 inhibitor [115,120]. Unfortunately, ATRis are ineffective in combination with subclinical levels of hydroxyurea or gemcitabine. Conversely, the combinations of CHK1i and subclinical dose of hydroxyurea or gemcitabine not only had little normal tissue toxicity [121,122]. By using subclinical doses of replication stress promoters’ gemcitabine and hy-droxyurea in combination of CHK1i and hydroxyurea [115,120,121]. Although, ATRi shows to have little normal tissue toxicity even in normally chemo-sensitive tissue such as immune cells, subclinical dosages are ineffective in these combinations. In preclinical models of melanoma and small cell lung cancer, combination of low dose hydroxyurea or gemcitabine with CHK1i trigger proinflam-matory cytokine and chemokine expression and enhance both innate and adaptive immune cell tumor infiltration and anti-tumor responses [115,120]. The immune responses triggered by these combinations differed depending on the cancer type, and ICB enhanced the immune response only in the small cell lung cancer models suggesting the immunosuppressive pathway differed between cancer types.

## 6. Use of SMIs Which Induce Epigenetic Changes

Epigenetic alterations contribute to carcinogenesis and significantly influence T and NK cell activation, differentiation, and function [123]. Therefore, strategic repurposing of epigenetically targeted drugs to boost immune cell function whilst suppressing pro-oncogenic signals could enhance clinical response to ICB mAbs [123]. Drugs targeting epigenetic alteration inhibit DNA methyltransferases (DNMTs), DNA demethylases, histone methyltransferases (HMTs), histone demethylases (HDMs) and other relevant enzymes involved in gene expression modulation [124,125,126,127,128]. In a murine B16-gp33 model, HDACi MS-275 induced NOS2 (Nitric Oxide Synthase 2)/Reactive Oxygen species (ROS) secretion and activated pro-inflammatory gene signatures which reduced the immunosuppressive function of tumor-infiltrating myeloid cells, by inducing their cell death in an IFN-γR/STAT1 signalling dependent manner [129].

Selective inhibition of Enhancer of Zeste Homolog 2 (EZH2), using CPI-1205, in a murine MC-38 cancer model disrupted the immunosuppressive function of tumor infiltrating T_regs,_ skewing their response towards a more pro-inflammatory phenotype. Effector CD4^+^ and CD8^+^ T cell numbers increased within the TME leading to tumor elimination [130]. Ghosh et al. [131] demonstrated that chemical inhibition of Cyclic adenosine monophosphate response element Binding Protein (CBP/EP300) bromodomain, using a series of laboratory synthesised inhibitors, led to the blockage of T_reg_ immunosuppressive function due to reduced FOXP3 acetylation which resulted in its degradation.

Bromodomain (BRD) and extra-terminal motif (BET) proteins inhibitors (BETi) have been shown to regulate the presentation and generation of neo-antigens, expression of immune checkpoints molecules, secretion of cytokines, and the activation of immune cells in several murine and human cancer settings [132]. Mechanistically, the BET family (BRD2, BRD3, BRD4, and BRDT) transcriptionally controls a range of proinflammatory and immunoregulatory genes by recognizing acetylated histones (mainly H3 and H4) and recreating necessary transcription factors and promote phosphorylation of RNA polymerase to the chromatin site [133]. BRD4 restores anti-tumor immune responses following chemical inhibition with small-molecule bromodomain inhibitor JQ1, by down regulating PD-L1 expression in a MYC dependent manner in multiple myeloma [134]. To date, JQ1 has been shown to downregulate the BRD4-MYC axis across several epithelial cancers, in preclinical and clinical studies [135]. Downregulation of the BRD4-MYC transcription axis using JQ1 resulted in boosting of stem cell–like and central memory CD8^+^ T cells responses that enhanced antitumor immunity in mouse models of epithelial ovarian cancer [136]. Similarly, BRD4 inhibition led to expression of proinflammatory genes such as Baculoviral IAP Repeat Containing 2 & 3 (BIRC2 and BIRC3), which in turn led to tumor necrosis factor (TNF) production triggering apoptosis in preclinical colon cancer models, boosting anti-tumour immunity [137]. Thus, combination of these drugs with ICB mAbs could prove effective against aggressive solid tumors, although optimisation of drug combinations in animal studies will be required.

## 7. SMIs Paving Way for Cytotoxic Lymphocytes to Transform into Super Killers

SMIs have been developed which inhibit immune suppressive mechanisms whilst activating innate and/or adaptive immune cell pathways. These chemical therapies have advantages compared to biological therapies (antibody and cell therapies) such as lower manufacturing and administrative costs. A relevant example are two small molecule inhibitors from Curis biopharmaceutical, phase-I trial CA-170 (antagonizes VISTA and PD-L1) (NCT02812875) and CA-327 (antagonizes TIM-3 and PD-L1) [138]. In contrast to ICB mAbs, these drugs can simultaneously antagonize multiple immune checkpoint receptors, increasing their potential to prevent tumor immune escape [139].

Toll-like receptors (TLRs) trigger innate immune responses by recognising pathogen-associated antigens. TLR agonists, and particularly TLR7/8 agonists, are potential immuno-oncologic therapeutic targets [139]. Imiquimod and derivative imidazoquinolines (resiquimod, 852A, 852A and VTX-2337) have been developed for systemic delivery and are currently under clinical trial [140,141,142,143]. These TLR agonists synergise with interferons (type I or II) and induce reprogramming of M2 immune-suppressive macrophages into M1 proinflammatory type [144,145,146]. TLR5 agonist entolimod induced NK-cell-dependent activation of DCs which resulted in stimulation of CD8^+^ T cells, triggering durable memory against aggressive colon and mammary metastatic mouse models [147].

N-formyl-kynurenine is a potent endogenous inhibitor of T cell activation produced by catabolism of tryptophan by heme-containing dioxygenase enzyme called IDO (indoleamine 2,3-dioxygenase) and helps tumor cells to evade immunosurveillance. Kynurenine metabolic pathway upregulation results in downregulation of tryptophan uptake as a consequence of which effector T cells function is reduced. Tryptophan is critical for TCR activation and hence is important in promoting antigen recognition. However, absence of tryptophan promotes T_reg_ function by activating aryl hydrocarbon receptor activation enabling tumor evasion. IDO is thus an important target in immune-oncology. IDO inhibitors (e.g., epacadostat) reduce tumor growth and promote the proliferation of CD8^+^ T cells and NK cells in human peripheral blood mononuclear cells (PBMCs) ex vivo and are currently under clinical trial (Table 2) [148].

Adenosine triphosphate (ATP) catabolism mediates immunosuppression, through inducing expression of CD39 and CD73, which regulate growth and metastasis of tumor cells. Tumor cells dephosphorylate ATP with the help of CD39 and CD73 to produce adenosine, which interacts with adenosine receptors A2aR and A2bR on cytotoxic lymphocytes and suppresses cytolysis [149]. Free ATP molecules are recognised as “danger” signals by the immune system and are known to activate the nucleotide-binding oligomerization domain (NLRP3) inflammasome in DCs and induce IL1-β, promoting an inflammatory response in cancer. SMIs against CD39 (ARL6715), CD73 (AMPCP) and adenosine receptors (CPI-444 inhibiting A2AR) have been shown to promote robust cytotoxic CD8^+^ T cell responses [150,151,152]. Collectively, these studies highlight the potential of small molecule-based immune therapies to “super activate” or prevent immune exhaustion which in turn enhances tumor killing.

## 8. Utilising SMIs to Induce Immunogenic Cell Death

Chronic exposure of damage-associated molecular patterns (DAMPs) in the TME can activate or suppress key multiple cellular pathways among cancer cells such as Caspase 3 or PIK3CA which results in immunogenic cell death (ICD) by necroptosis, ferroptosis or pyroptosis [153,154]. The release of DAMPs can be observed upon exposure to chemotherapeutic drugs, on-colytic viruses, physicochemical therapies, photodynamic therapy, and radiotherapy. An adaptive immune response can thus be triggered, initiating effector cytotoxic T cell function, and eliciting immunological memory by exposing [155]. When cells undergo ICD, there is a characteristic release of adenosine triphosphate (ATP) and high mobility group box 1 (HMGB1)) that leads to the activation of type I IFN responses and release of pro-inflammatory chemokines/cytokines (i.e., IL-1 and IL-18) [156,157,158]. As a result, immune cells can be recruited to the tumor, including cross-primed CD8^+^ T cells, due to availability of rich source of immunogens [159,160].

When cancer cells undergo cell death by necrosis, tumor cell DNA is released and detected by cGAS in APCs. cGAS is responsible for the production of Cyclic guanosine monophosphate–adenosine monophosphate (cGAMP) which binds to and activated STING [161]. Consequently, it activates ICD through activation of NF-κB (nuclear factor kappa B) pathway. This pathway results in production of IFNs and pro-inflammatory cytokines, promoting recruitment and activation of T cells [107]. In a preclinical pancreatic mouse model, STING agonist DMXAA reshaped the archi-tecture of the TME enabling more infiltration of activated cytotoxic T cells while re-ducing Tregs numbers. Additionally, DMXAA induced high expression of costimulatory molecules in cross-presenting DCs which resulted in reprogramming M2 into M1 macrophages [162]. Upregulation of anti-apoptotic proteins such as B-cell lymphoma-2 (BCL-2) and its homologues Bcl-xL and Bcl-w protect against tumour cell apoptosis by inhibiting mi-tochondrial outer membrane permeabilization. Navitoclax, a BCL-2 inhibitor, has been shown to reduce immune suppression, and proliferation and survival of cancer associ-ated fibroblasts (CAFs). CAF facilitate downregulation of ICD by suppressing release of ATP and HMGB-1 when exposed to radiation or chemotherapy which can induce resistance to ICD [163].In addition, CAFs restrict CD8^+^ T-cell infiltration which imposes immunologically cold TME leading to insensitivity towards ICB treatment in syngeneic breast cancer mouse model [164]. Hence, eliminating CAF population using navitoclax could potentially boost the efficacy of ICB.

To date, multiple chemo-drugs such as doxorubicin, mitoxantrone, oxaliplatin, and bortezomib, have been demonstrated to effectively induce tumor cell death. Artemisinin (ART)—a clinically approved anti-malarial drug—has been shown to have cytotoxic properties against tumor cells resulting in immune mediated cell death [165,166]. In an ex vivo experiment using an endometrial carcinoma cell line, ART upregulated the expression of immunosuppressive molecules such as CD155 (expressed on tumor cells) whilst downregulating TIGIT on NK cells which overall enhanced cytotoxicity when tumor and NK cells were cocultured [167].

Azacytidine and romidepsin (FDA-approved drugs) in combination with IFNα2 (ARI) have been shown to induce ICD in colorectal cancers cells in vitro, which in turn resulted in DCs stimulation due to upregulation of IFN. Increased DCs trafficking facilitated T cell cross-priming in tumor draining lymph nodes in a syngeneic colon cancer mouse model [168,169]. In a recent study, Zhang et al. [170] compared 4 SMIs (bortezomib and obatoclax mesylate vs. BI 2536 (BI) and (S)-(+)-camptothecin (CPT)). They demonstrated that BI and CPT triggered immune mediated cell death (pyroptosis) in syngeneic colon cancer mouse model leading to a greater CD8^+^ T cell accumulation at the tumor site compared to bortezomib or obatoclax mesylate which, conversely, did not induce ICD. Apurinic/apyrimidinic endonuclease 1 (APE1) inhibitor NO.0449-0145 has been shown to induce ICD in NSCLC preclinical models justifying investigation of the efficacy of this inhibitor in boosting anti-tumour immunity [171]. Cancer therapies that induce ICD should demonstrate enhanced effectiveness when combined with ICB mAbs, and it is likely that further promising combinatory therapies will be developed soon.

## 9. Future Perspective

The clinical efficacy and durability of ICB-based cancer immunotherapy has revolutionised the way solid tumors are treated and managed over the last decade. However, increasing the efficacy of ICB in a broader patient cohort continues to be challenging. With advancing technology, it is now understood that inherent oncogenic properties of cancer cells dictate the TME to alter the immune architecture. Targeting oncogenic signalling—especially the RTK signalling cascade—with SMIs in combination with ICB mAbs has gained traction and a wide range of RTKi are currently in phase III/IV clinical trials (currently recruiting or published). Several of these have shown significantly better clinical efficacy compared to ICB alone (Table 1 and Table 2). However, over the last decade, other SMIs known to inhibit oncogenic properties such as deregulated cell cycle, abnormal DNA damage regulation, epigenetic aberrations, metabolic abnormality, upregulated immune evasion molecules and suppressors of cell death mechanism have been found to modulate anti-tumor immunity in preclinical syngeneic epithelial cancer models (Figure 1). The major areas of research focus should be to: (i) elaborate on tissue-specific oncogene-related immune effects; (ii) delineate and functionally validate biomarkers which can predict response and resistance to oncogene targeting; (iii) generate and characterise highly dependable animal models to mimic human immune response to tumors (i.e., humanised mouse models) and (iv) develop multiplexed assays to incorporate immune and tumor intrinsic molecular changes in response to combination therapy. In addition, a comprehensive understanding of the TME architecture with spatial orientation of immune cells and their interaction with the cancer cells should be carefully examined for selecting the most ideal drugs for combination therapy of cancer.

Intermittent dosing of these SMIs which would accommodate treatment-free interval for the administration of ICB mAbs to potentiate high antigen expression and cytotoxic T cell infiltration is currently needed to avoid drug resistance and maximize treatment efficacy. As such optimization of these potential combinations should be investigated both at preclinical and clinical level including designing appropriate treatment schedules to attain enhanced anticancer immunosurveillance with minimum risk of toxicities. Using SMIs to induce an immunomodulatory effect in conjunction with making the hostile TME favourable to an anti-tumor response, provides a strong rationale for their combination with ICB mAbs. Combination therapy has the potential to synergistically inhibit malignant cells alongside augmenting the immune recognition and elimination of the tumor. Furthermore, with the ability to induce long-term antitumor memory, combination therapy may lead to greater rates of cure. Target therapies in combination with ICB mAbs might prove to be “game-changing” for patients with aggressive disease in the near future.

## Figures and Tables

**Figure 1 cancers-14-06150-f001:**
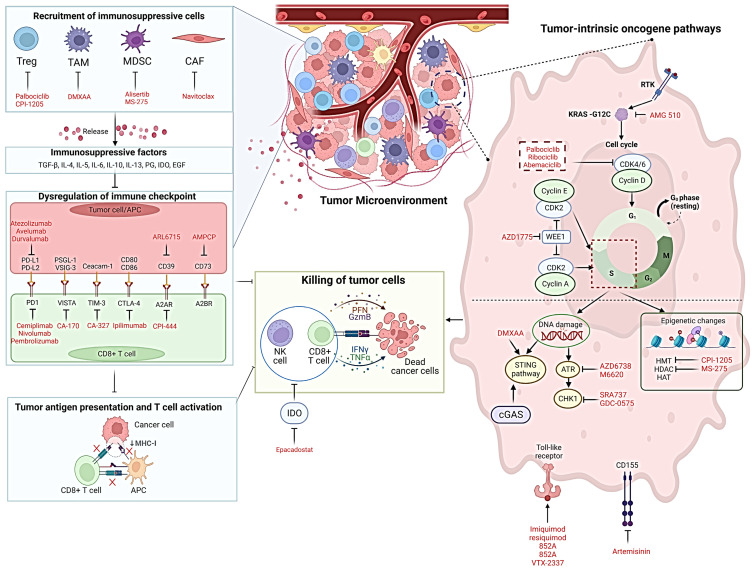
Tackling immunosuppression imposed by hostile TME using Small Molecule Inhibitors (SMIs) to improve ICB. The TME recruits immunosuppressive cells, for example, Treg, TAM, MDSC, and CAF. Those inhibitory cells release immunosuppressive factors (e.g., TGF-β, IL-4, IL-5, IL-6, IL-10, IL-13, PG, IDO, and EGF), and further cause the dysregulation of immune checkpoints, inhibition of tumor antigen presentation, and suppression of T cell activation. Among the inhibitors for immune checkpoints, atezolizumab, avelumab, and durvalumab target PD-L1, whereas semiplimab, nivolumab, and pembrolizumab target PD-1. There are also inhibitors for VISTA (CA-170), TIM-3 (CA-327), CTLA-4 (Ipilimumab), A2AR (CPI-444), CD39 (ARL6715), and CD73 (AMPCP). Such immunosuppressive environment suppresses the killing of tumor cells by CD8^+^ T cells and NK cells, enabling immune evasion. The inhibitors currently in use for inhibiting the shown intrinsic oncogenic signals (highlighted in red) have been shown to modulate immune response by overcoming immunosuppression. Therefore, these SMIs can be a powerful to enhance the clinical efficacy of ICB. Abbreviations: TME, tumor microenvironment; SMIs, small molecule inhibitors; Treg, regulatory T cells; TAM, tumor associated macrophages; MDSC, myeloid-derived suppressor cells; CAF, carcinoma associated fibroblasts; APC, antigen-presenting cell; HMT, methyltransferases; HDAC, histone deacetylases; HAT, histone acetyltransferases. Figure generated with BioRender^®^.

**Table 1 cancers-14-06150-t001:** Selected immunotherapy clinical trials, over the last ten years, which demonstrated no benefit. Abbreviations: HR—hazard ratio (Hazard ratio is the ratio of hazard rates between two different treatment groups. A hazard ratio of 1 indicates there is no difference in hazard rates between the two groups. Hazard ratios below 1 indicate that the treatment might be favorable). NSCLC—non-small cell lung cancer, OS—overall survival, PD-1—programmed death 1, PFS—progression free survival, SCLC—small cell lung cancer, SOC—standard of care.

NCT	ICB	Trial Arms	Population	Size	Results	Ref.
NCT00861614	Ipilimumab (CTLA-4)	Ipilimumab vs. placebo	Castration resistant prostate cancer with previous treatment with docetaxel	988	Median OS was 11.2 months (95% CI 9.5–12.7) with ipilimumab and 10.0 months (8.3–11) with placebo (hazard ratio [HR] 0.85, 0.72–1.00; *p* = 0.053)	[13]
NCT01057810	Ipilimumab (CTLA-4)	Ipilimumab vs. placebo	Castration-resistant prostate cancer—asymptomatic or minimally symptomatic with metastatic chemotherapy-naive	400	Median OS was 28.7 months (95% CI, 24.5 to 32.5 months) in the ipilimumab arm versus 29.7 months (95% CI, 26.1 to 34.2 months) in the placebo arm (hazard ratio, 1.11; 95.87% CI, 0.88 to 1.39; *p* = 0.3667)	[14]
NCT02617589	Nivolumab (PD-1)	Nivolumab vs. temozolomide	Newly diagnosed MGMT-unmethylated Glioblastoma	560	Press release—did not meet primary end points of OS or PFS.	[15]
NCT02017717	Nivolumab (PD-1)	Nivolumab vs. bevacizumab	Grade IV Glioblastoma	529	median OS (mOS) was comparable between groups: nivolumab, 9.8 months (95% CI, 8.2–11.8); bevacizumab, 10.0 months (95% CI, 9.0–11.8); HR, 1.04 (95% CI, 0.83–1.30); *p* = 0.76.	[16]
NCT02991482	Pembrolizumab (PD-1)	Pembrolizumab vs. SOC	Advanced malignant mesothelioma previously treated with platinum-based chemotherapy	144	No difference in OS was detected between groups (HR = 1.12, 95% CI: 0.74–1.69; *p* = 0.59)	[17]
NCT02555657	Pembrolizumab (PD-1)	Pembrolizumab vs. SOC	Metastatic triple negative breast cancer, previous treatment with two systemic therapies	622	In the overall population, median overall survival was 9.9 months (95% CI 8.3–11.4) for the pembrolizumab group and 10.8 months (9.1–12.6) for the chemotherapy group (HR 0.97 [95% CI 0.82–1.15]).	[18]
NCT02370498	Pembrolizumab (PD-1)	Pembrolizumab vs. SOC	Advanced gastric/gastroesophageal junction adenocarcinoma progressive after platinum-based chemotherapy	592	Median overall survival was 9.1 months (95% CI 6.2–10.7) with pembrolizumab and 8.3 months (7.6–9.0) with paclitaxel (hazard ratio [HR] 0.82, 95% CI 0.66–1.03; one-sided *p* = 0.0421).	[19]
NCT02494583	Pembrolizumab (PD-1)	Pembrolizumab vs. pembrolizumab plus SOC vs. SOC	Advanced Gastric or Gastroesophageal Junction Adenocarcinoma—first-Line Monotherapy and Combination Therapy	763	Pembrolizumab plus chemotherapy was not superior to chemotherapy for OS in patients with CPS of 1 or greater (12.5 vs. 11.1 months; HR, 0.85; 95% CI, 0.70–1.03; *p* = 0.05) or CPS of 10 or greater (12.3 vs. 10.8 months; HR, 0.85; 95% CI, 0.62–1.17; *p* = 0.16)	[20]
NCT02853305	Pembrolizumab (PD-1)	Pembrolizumab vs. pembrolizumab plus SOC vs. SOC	Advanced or metastatic urothelial carcinoma with no previous systemic therapy	1010	Pembrolizumab plus chemotherapy versus chemotherapy did not significantly improve overall survival, with a median overall survival of 17.0 months (14.5–19.5) in the pembrolizumab plus chemotherapy group versus 14.3 months (12.3–16.7) in the chemotherapy group (0.86, 0.72–1.02; *p* = 0.0407).	[21]
NCT02702401	Pembrolizumab (PD-1)	Pembrolizumab vs. placebo	Advanced hepatocellular carcinoma previously systemically treated	413	OS and PFS did not reach statistical significance per specified criteria. Median OS was 13.9 months (95% CI, 11.6 to 16.0 months) for pembrolizumab versus 10.6 months (95% CI, 8.3 to 13.5 months) for placebo (hazard ratio [HR], 0.781	[22]
NCT02551159	Durvalumab (PD-1)	Durvalumab vs. SOC	Recurrent/metastatic head neck squamous cell carcinoma—first line with high PD-1 expression	823	Press release—did not meet the primary endpoint of improving overall survival (OS) versus the EXTREME treatment regimen (chemotherapy plus cetuximab)	[23]
NCT02952586	Avelumab (PD-L1)	Avelumab + SOC vs. SOC	Locally advanced head neck squamous cell carcinoma	697	Median progression-free survival was not reached (95% CI 16.9 months–not estimable) in the avelumab group and not reached (23.0 months–not estimable) in the placebo group (stratified hazard ratio 1.21 [95% CI 0.93–1.57] favouring the placebo group; one-sided *p* = 0.92).	[24]
NCT02542293	Durvalumab (PD-1) + Tremelimumab (CTLA-4)	Combination immunotherapy vs. SOC	Metastatic NSCLC—first line	953	Press release—did not meet primary endpoints	[25]
NCT02538666	Nivolumab (PD-1) and Ipilimumab (CTLA-4)	Combination immunotherapy vs. placebo	Extensive disease NSCLC—as maintenance therapy post platinum-based chemotherapy	1212	OS was not significantly prolonged with nivolumab plus ipilimumab versus placebo (hazard ratio [HR], 0.92; 95% CI, 0.75 to 1.12; *p* = 0.37; median, 9.2 v 9.6 months)	[26]
NCT02279732	Ipilimumab (CTLA-4)	Ipilimumab + SOC vs. placebo + SOC	Metastatic or recurrent squamous NSCLC	342	ClinicalTrail.Gov result posted. Recruitment stopped at 204 patients and primary end point not analysed.	[27]
NCT01285609	Ipilimumab (CTLA-4)	Ipilimumab + SOC vs. placebo + SOC	Metastatic or recurrent squamous NSCLC	1289	Median OS was 13.4 months for chemotherapy plus ipilimumab and 12.4 months for chemotherapy plus placebo (hazard ratio, 0.91; 95% CI, 0.77 to 1.07; *p* = 0.25).	[28]
NCT01450761	Ipilimumab (CTLA-4)	Ipilimumab + SOC vs. placebo + SOC	Newly diagnosed extensive-stage SCLC	1351	Median OS was 11.0 months for chemotherapy plus ipilimumab versus 10.9 months for chemotherapy plus placebo (hazard ratio, 0.94; 95% CI, 0.81 to 1.09; *p* = 0.3775).	[29]

**Table 2 cancers-14-06150-t002:** Summary of trials combining immunotherapy and small molecule inhibitors (published or currently under trial in last 5 years). The mechanisms of action of the drug are indicated in brackets. Abbreviations: AKT—protein kinase B, ALK—anaplastic lymphoma kinase, BRAF—B-Raf, CTLA-4—cytotoxic T-lymphocyte-associated protein 4, HCC—hepatocellular carcinoma, HDAC—Histone deacetylases, HR—hazard ratio, IDO1—indoleamine 2,3-dioxygenase 1, MEK—mitogen-activated protein kinase, NSCLC—non-small cell lung cancer, ORR—objective response rate, PARP—poly ADP ribose polymerase, PD-1—programmed death-1; PFS—progression free survival, SCC—squamous cell carcinoma, SCLC—small cell lung cancer, SOC—standard of care, TACE—transarterial chemoembolism, TKR—tyrosine kinase receptor.

NCT	Trial Name	ICB	SMI	Trial Arms	Population	Size	Status	Outcomes	Refs.
Published/Completed Trials
NCT03361865	ECHO-007	Pembrolizumab (PD-1)	Epacadostat (IDO1)	1. Pembrolizumab + Epacadostat2. Pembrolizumab	Cisplatin-ineligible advanced or metastatic urothelial Carcinoma	93	Completed, not published	Source—ClinicalTrials.Gov ORR 31.8(22.46 to 55.24) vs. 24.5(15.33 to 43.67)	[46]
NCT02752074	ECHO-301	Pembrolizumab (PD-1)	Epacadostat (IDO1)	1. Pembrolizumab + Epacadostat 2. Pembrolizumab	Unresectable or metastatic melanoma	706	Completed	No significant difference in PFS or OS	[47]
NCT03829332	LEAP-007	Pembrolizumab (PD-1)	Lenvatinib(TKR)	1. Pembrolizumab + lenvatinib + SOC 2. Pembrolizumab + SOC	Treatment-naïve, Metastatic NSCLC	623	Completed, not published	ClinicalTrials.Gov PFS 6.6 months (Combination) vs. 4.2 months (Pembrolizumab monotherapy) HR 0.78 (*p* = 0.006). No benefit to overall survival.	[48]
NCT03517449	KEYNOTE-775	Pembrolizumab (PD-1)	Lenvatinib (TKR)	1. Pembrolizumab + lenvatinib 2. SOC	Advanced, recurrent or metastatic endometrial cancer.	827	Completed	PFS combo 7.2 vs. SOC 3.8 months; hazard ratio, 0.56; 95% CI, 0.47 to 0.66; *p* < 0.001. OS 8.3 vs. 11.4 months; hazard ratio, 0.62; 95% CI, 0.51 to 0.75; *p* < 0.001	[49]
NCT02853331	KEYNOTE-426	Pembrolizumab(PD-1)	Axitinib (TKR)	1. Pembrolizumab + Axitinib 2. Sunitinib	First-line in Locally Advanced or Metastatic Renal Cell Carcinoma	861	Completed	PFS -15.1 months pembrolizumab + axitinib group vs. 11.1—month sunitinib group (HR for disease progression or death, 0.69; 95% CI, 0.57 to 0.84; *p* < 0.001	[50,51]
NCT02684006	JAVELIN Renal 101	Avelumab (PD-L1)	Sunitinib (TKR)	1. Avelumab + axitinib 2. Sunitinib	First-line in Locally Advanced Renal Cell Carcinoma	888	Completed	Median PFS l 13.8 months combination vs. 8.4 months monotherapy (hazard ratio, 0.69; 95% CI, 0.56 to 0.84; *p* < 0.001	[52]
NCT02788279	IMblaze370	Atezolizumab(PD-L1)	Cobimetinib(MEK)	1. Atezolizumab 2. Cobimetinib + Atezolizumab 3. Regorafenib	Previously Treated Unresectable Locally Advanced or Metastatic Colorectal Adenocarcinoma	363	Completed	Not significant difference. Median overall survival was 8.87 months with atezolizumab plus cobimetinib, 7, 10 months with atezolizumab, and 8.51 months with regorafenib; HR 1.00 for the combination versus regorafenib and HR 1.19 (*p* = 0.34) for atezolizumab versus regorafenib	[53]
NCT03141177	CheckMate 9ER	Nivolumab (PD-1)	Cabozantinib(TKR)	1. Nivolumab and Cabozantinib 2. Sunitinib 3. Nivolumab, Ipilimumab, Cabozantinib (discontinued)	First line Advanced or Metastatic Renal Cell Carcinoma	701	Completed	PFS 16.6 months (95% CI, 12.5 to 24.9) with nivolumab + cabozantinib vs. 8.3 months (95% CI, 7.0 to 9.7) sunitinib (HR 0.51; 95% CI, 0.41 to 0.64; *p* < 0.001). OS at 12 months 85.7% (95% CI, 81.3 to 89.1) with nivolumab + cabozantinib vs. 75.6% (95% CI, 70.5 to 80.0) with sunitinib (HR 0.60; 98.89% CI, 0.40 to 0.89; *p* = 0.001).	[54]
NCT03937219	COSMIC-313	Nivolumab (PD-1_ and Ipilimumab (CTLA-4)	Cabozantinib(TKR)	1. Cabozantinib + nivolumab + ipilimumab followed by cabozantinib + nivolumab 2. nivolumab + ipilimumab followed by nivolumab	First line Advanced or Metastatic Renal Cell Carcinoma of Intermediate or Poor Risk	840	Completed. Collecting OS data	press release/meeting abstract. Primary PFS endpoint (HR 0.73, 95% CI, 0.57–0.94; *p* = 0.013) in favour of combination	[55]
NCT03713593	LEAP-002	Pembrolizumab(PD-1)	Lenvatinib(TKR)	1. lenvatinib plus pembrolizumab 2. Lenvatinib + placebo	First-line Therapy for Advanced HCC	794	Completed	Press release—did not meet primary outcome measures	[56]
In Progress Trials (by Tumor Type)
NCT04335006		Carelizumab (PD-1)	Apatinib(TKR)	1. Carelizumab + Nab-paclitaxel + Apatinib 2. Carelizumab + Nab-paclitaxel 3. Nab-paclitaxel	Advanced or metastatic Triple Negative Breast Cancer	780	Recruiting	PFS	
NCT04177108		Atezolizumab (PD-L1)	Ipatasertib(AKT)	1. Paclitaxel, Atezolizumab and Ipatasertib 2. Paclitaxel, ipatasertib b 3. Paclitaxel	Locally Advanced Unresectable or Metastatic Triple-Negative Breast Cancer.	242	Active, not recruiting	PFS, OS	
NCT03740165	KEYLYNK-001	Pembrolizumab(PD-1)	Olaparib(PARP)	1. Pembrolizumab + Olaparib + SOC 2. Pembrolizumab + SOC 3. SOC	BRCA Non-mutated Advanced Epithelial Ovarian Cancer	1284	Active, not recruiting	PFS	
NCT05145218		TQB2450(PD-L1)	Anlotinib (TKR)	1. TQB2450 + Anlotinib 2. Paclitaxel	Recurrent platinum-resistant ovarian cancer	405	Recruiting	PFS, OS	
NCT03651206	ROCSAN	Dostarlimab (PD-1)	Niraparib(PARP)	1. Niraparib 2. Niraparib + TSR-042 (Dostarlimab) 3. SOC	Metastatic or Recurrent Endometrial or Ovarian Carcinosarcoma	196	Recruiting	RR, OS	
NCT03598270		Atezolizumab (PD-L1)	Niraparib(PARP)	1. SOC 2. SOC + Atezolizumab with maintaince atezolizumab + niraparib	Recurrent ovarian cancer	414	Active, not recruiting	PFS	
NCT03793166	PDGREEI	Nivolumab (PD-1)	Cabozantinib(TKR)	1. Nivolumab 2. Nivolumab + Cabozantinib	Metastatic clear cell renal cancer	1046	Recruiting	OS	
NCT04523272		TQB2450 (PD-L1)	Anlotinib(TKR)	1. TQB2450 + Anlotinib 2. Sunitinib	Locally advanced clear cell renal cancer	418	Recruiting	PFS	
NCT05219318	SPICI	PD-1/PD-L1 ICI	VEGFR-Tyrosine Kinase Inhibitor	1. Treatment pause post-12 months of therapy. 2. PD-1/PD-L1 inhibitor + TKI	Good or Intermediate Risk Metastatic Renal Cell Carcinoma	372	Not yet recruiting	PFS	
NCT04338269	CONTACT-03	Atezolizumab(PD-L1)	Cabozantinib(TKR)	1. Atezolizumab + cabazntinib 2. cabazantinib	Inoperable, Locally Advanced, or Metastatic Renal Cell Carcinoma	523	Active, not recruiting	PFS, OS	
NCT04987203		Nivolumab (PD-1)	Tivozanib (TKR)	1. Nivolumab + Tivozanib 2. Tivozanib	Locally advanced or metastatic Renal cell carcinoma-with progression following at least 6 weeks of treatment with ICI	326	Recruiting	PFS	
NCT03898180	LEAP-011	Pembrolizumab(PD-1)	Lenvatinib (TKR)	1. Pembrolizumab + Lenvatinib2. Pembrolizumab monotherapy3. Placebo + pembrolizumab	First-line Cisplatin-ineligible Participants with PDL1 expression. Ineligible for Platinum-containing Chemotherapy Urothelial Carcinoma	487	Active, not recruiting	PFS, OS	
NCT03834519	KEYLYNK-010	Pembrolizumab(PD-1)	Olaparib(PARP)	1. Pembrolizumab + Olaparib 2. Abiraterone + Prednisone or Enzalutamide	Metastatic Castration-resistant Prostate Cancer	793	Active, not recruiting	PFS, OS	
NCT03976375	LEAP-008	Pembrolizumab(PD-1)	Lenvatinib(TKR)	1. Pembrolizumab + Lenvatinib 2. Docetaxel 3. Lenvatinib monotherapy	Metastatic NSCLC	405	Active, not recruiting	OS, PFS	
NCT03178552		Atezolizumab(PD-L1)	Cobimetinib (MEK), Alectinib (ALK), Entrectinib (ROS1), Vemurafenib (BRAF), GDC-6036 (KRAS)	Multiple trial arms including different combinations	Advanced or metastatic NSCLC	1000	Recruiting	ORR	
NCT04471428		Atezolizumab(PD-L1)	Cabozantinib (TKR)	1. Atezolizuman + cabozantinib 2. Docetaxel	Metastatic NSCLC	366	Active, not recruiting	OS	
NCT04921358	SAFFRON-301:	Tislelizumab(PD-1)	Sitravatinib(TKR)	1. Tislelizumab + Sitravatinib 2. Docetaxel	Metastatic NSCLC	420	Recruiting	OS, PFS	
NCT03348904		Nivolumab(PD-1)	Epacadostat(IDO1)	1. Nivolumab + epacadostat + platnium 2. Platinum chemotherapy 3. Platinum + Nivolumab	Metastatic or recurrent NSCLC	2	Terminated early		
NCT04380636	KEYLYNK-012	Pembrolizumab(PD-1)	Olaparib (PARP)	1. pembrolizumab + chemoradiation → pembrolizumab + olaparib placebo 2. pembrolizumab + chemoradiation → pembrolizumab + olaparib 3. chemoradiation → durvalumab	Unresectable, locally advanced NSCLC	870	Recruiting	PFS, OS	
NCT03906071	SAPPHIRE	Nivolumab(PD-1)	Sitravatinib(TKR)	1. Nivolumab and Sitravatinib 2. Docetaxel	Advanced or metastatic NSCLC	532	Active, not recruiting	OS	
NCT03976362	KEYLYNK-008	Pembrolizumab(PD-1)	Olaparib(PARP)	1. Pembrolizumab + Carboplatin + Taxane + Maintenance Olaparib 2. Pembrolizumab + Carboplatin + Taxane + Maintenance placebo	First-line Metastatic NSCLC	857	Active, not recruiting	PFS, OS	
NCT03976323	KEYLYNK-006	Pembrolizumab (PD-1)	Olaparib(PARP)	1. Pembrolizumab + Pemetrexed + Platinum Therapy + Maintenance Olaparib 2. Pembrolizumab + Pemetrexed + Platinum Therapy + Maintenance Pemetrexed	First-line Metastatic NSCLC	1005	Active, not recruiting	PFS, OS	
NCT03829319	LEAP-006	Pembrolizumab(PD-1)	Lenvatinib(TKR)	1. Pembrolizumab + lenvatinib + SOC 2. Pembrolizumab + SOC	Metastatic Nonsquamous NSCLC	726	Active, not recruiting	Safety, PFS, OS	
NCT05042375		Camrelizumab(PD-1)	Famitinib(TKR)	1. camrelizumab + famitinib 2. pembrolizumab 3. camrelizumab	PD-L1-Positive Recurrent or Metastatic NSCLC	450	Not yet recruiting	PFS	
NCT05346952		TQB2450(PD-L1)	Anlotinib (TKR)	1. TQB2450 + carboplatin + pemetrexed 2. TQB2450 + Anlotinib + Pemetrexed	First-line Treatment on Patient with Advanced Non-squamous NSCLC	390	Recruiting	PFS, OS	
NCT05106335		Camrelizumab (PD-1)	Famitinib(TKR)	1. Camerlizumab + famitinib 2. famitinib 3. docetaxel	Advanced NSCLC	524	Recruiting	OS	
NCT04234607	ETER701	TQB2450 (PD-L1)	Anlotinib (TKR)	1. TQB2450 + Anlotinib + etoposide + carboplatin 2. Anlotinib + etoposide + carboplatin 3. etoposide + carboplatin	Extensive SCLC	738	Not yet recruiting	PFS, OS	
NCT04624204	KEYLYNK-013	Pembrolizumab (PD-1)	Olaparib(PARP)	1. Pembrolizumab + SOC 2. Pembrolizumab + Olaparib + SOC 3. SOC	Newly Diagnosed Treatment-Naïve Limited-Stage SCLC	672	Recruiting	PFS, OS	
NCT04674683		Nivolumab(PD-1)	HBI-8000(HDAC)	1. HBI-8000 + nivolumab2. Placebo + nivolumab	Unresectable or metastatic melanoma	480	Recruiting	ORR, PFS	
NCT03820986	LEAP-003	Pembrolizumab (PD-1)	Lenvatinib(TKR)	1. Lenvatinib + pembrolizumab2. Pembrolizumab + placebo	First-line in adults With Advance Melanoma	660	Active, not recruiting	PFS, OS	
NCT03813784		SHR-1210 (PD-1)	Apatinib(TKR)	1. SHR-1210 + Apatinib + SOC2. SOC3. SOC + SHR-1210	Advanced or metastatic gastric cancer	887	Active, not recruiting	OS	
NCT04949256	LEAP-014	Pembrolizumab (PD-1)	Lenvatinib (TKR)	1. Pembrolizumab + Lenvatinib + Chemotherapy 2. Pembrolizumab + Chemotherapy	First-line Metastatic Esophageal Carcinoma	862	Recruiting	Safety, PFS, OS	
NCT04662710	LEAP-015	Pembrolizumab (PD-1)	Lenvatinib(TKR)	1. Lenvatinib + Pembrolizumab + SOC2. SOC	First-line in Advanced/Metastatic Gastroesophageal Adenocarcinoma	790	Recruiting	PFS, OS	
NCT04879368	INTEGRATEIIb	Nivolumab(PD-1)	Regorafenib(TKR)	1. Nivolumab + regorafenib 2. SOC	Refractory Advanced Gastro-Oesophageal Cancer	450	Recruiting	OS	
NCT05049681		SHR-1210 (PD-1)	Apatinib(TKR)	1. SHR-1210 + Apatinib 2. SHR-1210	Locally advanced/unresectable, recurrence or metastatic esophegeal SCC	234	Not yet recruiting	OS	
NCT04776148	LEAP-17	Pembrolizumab (PD-1)	Lenvatinib(TKR)	1. lenvatinib + pembrolizumab2. SOC	Metastatic Colorectal Cancer	424	Active, not recruiting	OS	
NCT04669496		Toripalimab (PD-1)	Lenvatinib(TKR)	1. Neoadjuvant GEMOX + Lenvatinib + Toripalimab2. No neoadjuvant therapy	Resectable Intrahepatic Cholangiocarcinoma with High-risk Recurrence Factors	178	Recruiting	PFS	
NCT04246177	LEAP-012	Pembrolizumab (PD-1)	Lenvatinib(TKR)	1. Lenvatinib plus Pembrolizumab plus TACE2. Oral Placebo plus IV Placebo plus TACE	Incurable Locally Advanced HCC	950	Recruiting	PFS, OS	
NCT04523493		Toripalimab(PD-1)	Lenvatinib(TKR)	1. Toripalimab + Lenvatinib 2. Lenvatinib	First-line Therapy for Advanced HCC	519	Recruiting	PFS, OS

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
