# Peer review of "Repurposing of Commercially Existing Molecular Target Therapies to Boost the Clinical Efficacy of Immune Checkpoint Blockade"

_cancers, 2022, doi:10.3390/cancers14246150_

Round 1
Reviewer 1 Report
In this study, the authors reviewed combination strategy of immune checkpoint blockade and small molecular inhibitors, which attenuated tumor immunosuppression and enhanced ICB efficacy. While there will be some interest in the field for the investigation of immune checkpoint blockade therapy, the manuscript is lacking some important information to allow for publication.
1. We recommend the authors provided more information of immune checkpoint blockade therapy in the introduction, such as the classification and mechanisms of immune checkpoint inhibitors.
2. In the Figure 1, the authors illustrate some small molecular inhibitors improved the immune checkpoint blockade efficacy through the modulation of tumor cells. But the authors should also update Toll-like receptors agonists, IDO inhibitors and immunogenic cell death related drugs in this figure.
3. In this review the author summarized the combination strategy focused on PD-1/PD-L1 antibody. We recommend the authors updated more combination strategy related to anti-TIGIT antibody, anti-CD47 antibody and anti-LAG3 antibody.
Author Response
Reviewer #1
Estimated reviewer,
we would like to thank you for taking the time to review our work and making constructive comments and suggestions. We have taken them into account, and substantial changes have been made to increase the quality and readability of the manuscript.
In the revised manuscript, main text changes are highlighted in red. Figure 1 has been modified and additional pathways have been added as suggested.
Please, find below our individual responses to your comments:
- We recommend the authors provided more information of immune checkpoint blockade therapy in the introduction, such as the classification and mechanisms of immune checkpoint inhibitors.
we have added the description of the main classification and mechanism of action of immune checkpoint blockers, in the “1. Introduction” section, page 1, lines 33-40 and page 2, lines 49-76. We have also updated the information about the recently FDA-approved ICBs anti-LAG3 and anti-TIGIT (page 2, lines 49-52).
- In the Figure 1, the authors illustrate some small molecular inhibitors improved the immune checkpoint blockade efficacy through the modulation of tumor cells. But the authors should also update Toll-like receptors agonists, IDO inhibitors and immunogenic cell death related drugs in this figure.
We have now updated Figure 1 with the agonist/inhibitors against Toll-like receptors, IDO enzyme and immunogenic cell death pathways as suggested (Page 8).
- In this review the author summarized the combination strategy focused on PD-1/PD-L1 antibody. We recommend the authors updated more combination strategy related to anti-TIGIT antibody, anti-CD47 antibody and anti-LAG3 antibody.
Under section “3. Repurposing SMIs to improve efficacy of ICB”, page 10-11, lines 205-244, we have now included the available information about SMIs against TIGIT, CD47 and LAG3 and added the pre-clinical data of the combination therapy of check point SMI plus ICB mAbs (Page 10, lines 220-225).

Reviewer 2 Report
This is a very interesting and comprehensive review dealing with potential combinations enhancing the ICB response in tumors. It is known that an immunosuppressive TME reduces the potential of ICB in several tumors and the article highlights new potential combinations with compounds interfering with cell cycle, DDR. These have the potential to decrease the presence/activity of immunuppressive players and hence enhance responce.
The review is definitely of interest for readers of the journal.
As a minor point i would stress and discuss the new potential of KRAS specific inhibitor as enhancer of ICB response, as shown in recent papers in NSCLC. This is in my opinion an emerging area of great potential interest being NSCLC one of the tumor in which ICB response, although important can definitely be further enhanced
Author Response
Reviewer #2
Estimated reviewer,
we would like to thank you for taking the time to review our work and making constructive comments and suggestions. We have taken them into account, and substantial changes have been made to increase the quality and readability of the manuscript.
In the revised manuscript, main text changes are highlighted in red. Figure 1 has been modified and additional pathways have been added.
- Inclusion of information about KRAS specific inhibitor as enhancer of ICB response in NSCLC.
We have now added relevant information about available SIMs for targeting KRAS-G12C and their synergistic effect when combined with ICBs in NSCLC (Page 26, lines 339-353). Relevant clinical trials have been included and a more comprehensive review about the topic cited for reference to those readers who would like a deeper understanding of the topic (Tani et al, 2021, PMID: 33703985).